# Prevalence and determinants of anxiety, depression, and suicidal ideation among adolescents of Parbat District: A cross-sectional study

**Anjali P.C**[1]*, **Rajan Bhusal**[2,3], **Anjali Bhatt**[4], **Prabin Sharma**[1], **Shreesti Sharma**[2], **Suman Sharma**[1], **Krishna Prasad Sapkota**[5]

**1** Province Health Office Kushma Parbat, Ministry of Health, Gandaki Province, Nepal, **2** Hospital and Rehabilitation Center for Disabled Children, Banepa, Nepal, **3** B and B Hospital, Gwarko, Nepal, **4** Central Department of Public Health, Kathmandu, Nepal, **5** Department of Sociology and Gerontology, Miami University, Oxford, Ohio, United States of America

* anjaleepc6@gmail.com

## Abstract

Adolescence is a critical human developmental stage where rapid biological, psychological, and social transitions occur. These changes may increase susceptibility to mental health conditions, including anxiety, depression, and suicidal ideation. Moreover, such conditions are leading contributors to disability and mortality among youth worldwide. Despite this, limited research has addressed these issues in Nepal. This study examined the prevalence and associated factors of anxiety, depression, and suicidal ideation among high school adolescents in Parbat District, Nepal. A cross-sectional study was conducted in August 2024 among 304 students in grades 11 and 12. Validated Nepali-language instruments, including the GAD-7, PHQ-9, and SBQ-R, were used to assess anxiety, depression, and suicidal ideation, respectively. Multivariable logistic regression analyses identified factors associated with each condition, with results reported as adjusted odds ratios (aOR) and 95% confidence intervals (CI). The prevalence of anxiety, depression, and suicidal ideation was 36.5%, 17.4%, and 7.6%, respectively. Female adolescents exhibited higher odds of anxiety (aOR: 2.59; 95% CI: 1.30–5.19) and depression (aOR: 5.65; 95% CI: 1.82–17.53). Similarly, non-Hindu participants had higher odds of anxiety (aOR: 6.63; 95% CI: 1.96–22.42) and depression (aOR: 7.86; 95% CI: 1.94–31.85). Lower scholastic achievement was significantly associated with anxiety (aOR: 0.27, [0.09 − 0.81], p < 0.05) and depression (aOR: 0.13, [0.02 − 0.70], p < 0.01). Likewise, childhood trauma was associated with depression (aOR: 15.61, [3.18 − 76.67], p < 0.001) and suicidal ideation (aOR: 16.14, [1.89 − 138.15], p < 0.01). A prior history of mental health problems was strongly linked to suicidal ideation (aOR: 38.83; 95% CI: 4.71–320.26). These findings suggest a substantial mental health burden among adolescents. Early identification and targeted, school-based interventions are important,

**Data availability statement:** All relevant data are within the manuscript and its Supporting information files.

**Funding:** The author(s) received no specific funding for this work.

**Competing interests:** The authors have declared that no competing interests exist.

especially for vulnerable groups such as female adolescents and those with poor academic performance or adverse childhood experiences.

## Background

From the ages of 10–19, adolescence is a unique time of human development marked by significant physical, cognitive, and psychological changes [1]. As it has significant long-term consequences for individuals and society, it is a crucial period to establish trajectories of good health and wellbeing. Approximately 1.2 billion adolescents worldwide reside in low- and middle-income countries, which is approximately 90% of global adolescents' population and this number is expected to increase by 2050. Currently, suicide and various mental health challenges rank among the leading causes of mortality for adolescents, highlighting a critical health concern within this demographic [1]. In fact, one in seven adolescents experiences mental health disorders, contributing to 15% of the global disease burden in this age group [2]. During adolescence, rapid physical, emotional, and social changes combined with factors such as poverty, violence, or neglect can increase the risk of developing mental health issues [2]. These conditions frequently go unrecognized and untreated, leading to further complications, including risk-taking behaviors (e.g., substance use, violence), social isolation, discrimination, stigma, and educational challenges [2,3]. To enhance overall wellbeing and reduce associated long-term health consequences, it is crucial to identify and address mental health challenges in adolescents. Prior studies have shown that anxiety, social relationships, substance abuse, and academic stress are major factors leading to suicidal behavior among adolescents, especially late adolescents [4–6].

The 2019 Global Burden of Disease study found a 3.6% prevalence of major depressive disorder in Nepal, and was higher than the regional average of 2.6% in South Asia and the global average of 2.5% [7]. Furthermore, a World Health Organization (WHO) survey in 2017 reported that Nepal had the highest rates of suicidal ideation (14%) and suicide attempts (10%) among adolescents in WHO South East Asia region [8]. Similarly, various community-based research carried out among high school students across different areas of Nepal also has shown a wide range of prevalence related to symptoms of anxiety and depression (anxiety: 23%–45%; depression: 13%–40%) [9–13]. The National Mental Health Survey (NMHS)2020 also reported 3.9% of adolescents in Nepal currently experiencing suicidal thoughts, with 0.7% having attempted suicide in their lifetime [14]. Additionally, indicators of mental health distress among adolescents revealed that 5% experienced anxiety and 7% reported feelings of loneliness [15]. Previous research shows that female adolescents are more likely to experience anxiety and depression than males, both globally and in Nepal. This disparity is influenced not only by biological factors but also by social and psychological mechanisms. Gender role socialization, lower perceived social support, ruminative coping, and differences in stress appraisal increase females' vulnerability to internalizing disorders, potentially explaining the higher risk of

depression compared to anxiety. The NHMS further showed the prevalence of mental disorders at 5.1% among adolescents in Gandaki province [14]. In the Parbat district of Gandaki province, TPO Nepal's recent screenings among 406 adolescents revealed that 16.5% of adolescents experience depression, 23.5% face stress, 9.3% suffer from severe mental disorders, and 6.4% are affected by alcohol use disorder [16]. In another screening survey with a larger sample of 2,923adolescents in Parbat, the prevalence rates were 12.9% for depression, 19.2% for stress, 5.3% for severe mental disorders, and 6.5% for alcohol use disorder, with higher rates observed in Kushma municipality [16]. Moreover, a survey conducted by the Nepal Police revealed that the prevalence of suicide among individuals aged 10–18 years is 14.19%, with 376 recorded suicides of 10-year-olds in the Parbat district [17]. Even with governmental attempts to tackle these problems, the mental health crisis in adolescents keeps escalating, and community awareness remains inadequate [18]. The issue of mental health is getting worse in Nepal, and adolescents are disproportionately affected [19–21]. Few studies have been carried out in Nepal to investigate the mental health of adolescents and the factors influencing it in this age group, particularly in the Parbat district of Nepal. This absence of research highlights a concerning deficit in comprehending the mental health issues confronting Nepali adolescents, especially considering their susceptibility to different socio-environmental influences. Most studies in Nepal have focused on higher secondary school students (grades 11 and 12), which limits the generalizability of findings to the broader adolescent population aged 10–19 [9,22].

This study aims to bridge this gap by investigating the prevalence of anxiety, depression, and suicidal ideation among adolescents in the Parbat district. This specific age group was selected because it represents late adolescence, a critical developmental period characterized by heightened vulnerability to mental health challenges and emotional difficulties. By generating baseline data, this research will help stakeholders better understand the mental health landscape of adolescents, enabling them to design targeted interventions and implement programs that effectively promote adolescent mental wellbeing at the local level.

## Methodology

### Ethical statement

This study complied with the Declaration of Helsinki and the National Ethical Guidelines for Health Research in Nepal, 2022. Ethical approval was obtained from the Ethical Review Board of the Nepal Health Research Council (Ref. No: 29, dated 21st July 2024). For adolescents under 18 years, written informed consent was obtained from the school principal or a designated teacher, as direct parental consent was not feasible. In addition, assent was obtained from the students themselves after providing clear information about the study, its voluntary nature, and the absence of foreseeable harm.

Confidentiality and anonymity were maintained throughout data collection, analysis, and reporting. Adolescents who screened positive for depression, anxiety, or suicidal ideation were sensitively counseled and referred, with the knowledge of school authorities, to appropriate mental health services for further evaluation and support. Participation remained entirely voluntary, with the option to withdraw at any time without any consequences.

### Study design and setting

A cross-sectional study was conducted in Kushma Municipality of the Parbat District, Gandaki Province, Nepal (Fig 1). The Parbat district with a diverse adolescent population from different ethnic backgrounds, and Kushma municipality, the fastest growing urban setting in the district, was chosen for the study, considering the fact of increased risk of mental health issues in the urban settings. The Kushma Municipality, located at 28.2358° N latitude and 83.6880° E longitude, is the largest municipality in the Parbat District and serves as district's prime urban center, including the headquarters of the district. The area draws students from across the district, representing a wide range of socio-economic backgrounds and providing a comprehensive view of the adolescent population in the area.

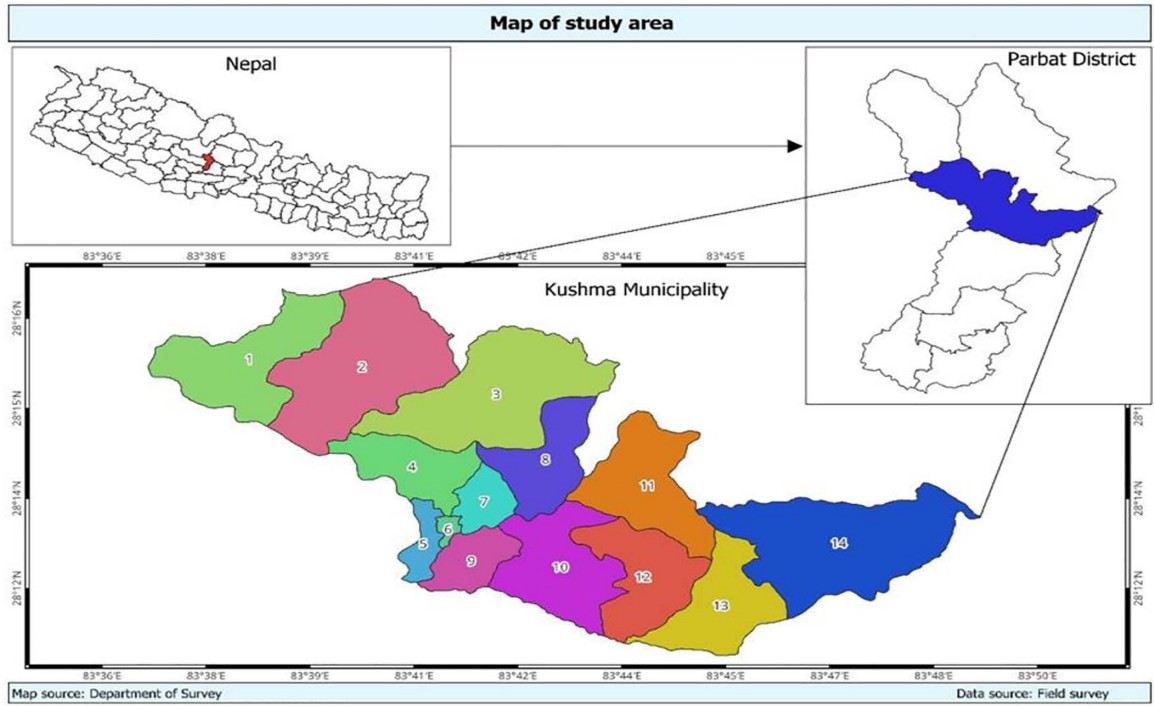

**Fig 1. Study area map of Parbat District, Nepal.** Base map source: Department of Survey, Government of Nepal. https://dos.gov.np/.

### Study participants and sampling design

A comprehensive list of all eligible adolescents was compiled from 10 higher secondary schools in Kushma Municipality, yielding a sampling frame of 1,078 students [23]. The sampling frame was obtained from the Kushma Municipality Education Section, which included adolescents aged 16–19 years enrolled in grades 11 and 12 from both government and private schools. Both government and private schools were included to ensure diversity in socioeconomic backgrounds and educational settings, allowing for a better understanding of adolescent mental health.

To obtain a representative sample, proportionate sampling was first used to allocate the number of participants from each school based on the total student population obtained from the Kushma Municipality Education Section through the Integrated Educational Management Information System (IEMIS). Following school approval, systematic random sampling was implemented by selecting a random starting point in each attendance register and choosing every $n^{th}$ student thereafter. This procedure ensured that all eligible students had an equal chance of inclusion, providing a representative sample of school-going adolescents in the municipality.

Inclusion criteria were students who provided informed consent and were present on the day of data collection. While this sampling approach ensures representativeness of school-going adolescents, the findings may not generalize to adolescents who are out of school or residing outside Kushma Municipality.

This sample size was determined using the formula $n = n = Z^2 \times p \times (1 - p)/d$, where p represented the expected proportion of the target condition and p = 40.6% was taken as a prevalence of depression/anxiety from a similar adolescent population in western Nepal which is socio-culturally and geographically comparable to Parbat district [23]. Furthermore, d was the margin of error (set at 5%), and Z was the standard normal variate (1.96 for a 95% confidence interval). After adjusting for a 10% non-response rate, the final sample size was 304. This sample size was considered adequate to detect meaningful associations between risk factors and mental health outcomes in the study population.

## Data collection

Data was collected using a structured questionnaire developed through rigorous literature review and was finalized after consultation with experts. The questionnaire was originally developed in English language and later was translated into Nepali language ensuring translation validity by team members. The questionnaire collected information about depression, anxiety, suicidal thoughts, socio-demographic, scholastic achievements, history of trauma and mental issues, among others. Depression, anxiety and suicidal ideation among adolescents was measured using standardized tools including Generalized Anxiety Disorder Scale (GAD-7), Patient Health Questionnaire (PHQ-9), and Suicidal Behaviors Questionnaire-Revised (SBQ-R), respectively. The contextual relevance of standardized tool was further ensured through review of studies using these tools in Nepal and further through expert consultation. The questionnaire was distributed to students who were randomly selected. Each student completed the questionnaire individually under the supervision of the researcher to ensure accuracy and consistency.

To verify clarity, relevance, and reliability, the questionnaire was pretested with 30 participants (10% of the total sample size of 304) on in Phalewas Municipality, located within the same district. Although Phalewas differs slightly in urbanization and school infrastructure compared to Kushma, both municipalities are demographically similar in terms of age distribution, ethnicity, and school enrollment patterns. Therefore, the pretest results were considered valid for refining the data collection tools and procedures for the main study in Kushma Municipality. Based on the feedback obtained from the pretest, only minor adjustments were made in consultation with the research team. For example, an additional response option— "foreign worker"—was added under the employment category to better capture the local context. These adjustments enhanced the overall quality and reliability of the questionnaire before its final implementation. Data collection took place from 28th July to 4th August 2024. Questionnaires were distributed to systematically randomly selected students, who completed them individually under researcher's supervision to ensure accuracy and consistency. The completed questionnaires were entered into Microsoft Excel and then imported into IBM SPSS version 26 for analysis.

## Measures

**Dependent variables.** The dependent variables in this study were anxiety, depression, and suicidal ideation, which are key indicators of adolescent mental health. These variables were assessed using reliable and valid standardized tools, which were translated into Nepali for the study [24].

Anxiety was measured using GAD-7 scale, a self-report questionnaire comprising seven items that assess the severity of anxiety symptoms over the past two weeks. Each item is rated on a 4-point Likert scale ranging from 0 (not at all) to 3 (nearly every day), yielding a total score between 0 and 21. Higher scores indicate more severe anxiety symptoms. The GAD-7 has a sensitivity of 89% and specificity of 82% for detecting generalized anxiety disorder. In this study, the internal consistency of the GAD-7 was excellent (Cronbach's $\alpha = 0.927$), indicating high reliability [25].

Depression was assessed using PHQ-9, a nine-item self-report measure evaluating the presence and severity of depressive symptoms over the preceding two weeks, with a sensitivity of 88% and specificity of 85% for detecting major depressive disorder [26]. Each item is scored from 0 (not at all) to 3 (nearly every day), resulting in a total score ranging from 0 to 27. Higher scores correspond to increased severity of depression. The PHQ-9 has been validated for use with adolescents, demonstrating strong psychometric properties in various studies [27]. In this study, internal consistency was high, with Cronbach's alpha values of 0.924 for PHQ-9, indicating excellent reliability.

Suicidal ideation was evaluated using the SBQ-R, which assesses past suicidal behavior and the risk of future suicides. The SBQ-R consists of four items assessing different dimensions of suicidal thoughts and behaviors and is measured in a Likert-scale score of 0 (never) to 4 (very often). The total score ranges from 3 to 18, with higher scores indicating a greater risk of suicide. The SBQ-R has shown strong reliability and validity in adolescent populations and is considered an appropriate tool for assessing suicide risk. It demonstrated a sensitivity of 92% and specificity of 88% for predicting suicide risk [28]. In the current study, internal consistency was strong (Cronbach's $\alpha = 0.88$) for SBQ-R.

All three conditions were categorized as a binary variable. Participants scoring 10 or above on the GAD-7 or PHQ-9 were categorized as "Yes" (clinically significant anxiety or depression), while those scoring below 10 were categorized as "No." For the SBQ-R, a score of 7 or higher indicated "Yes" (at risk of suicidal behavior), and scores below 7 were categorized as "No."[1][2][3]

**Independent variables.** The independent variables in this study were gathered through a structured questionnaire focusing on demographic, familial, social, academic, and behavioral factors that may influence adolescents' mental health outcomes. Age (in completed years), sex (male/female), and ethnicity (Dalit/Janajati/Brahmin–Chhetri) were recorded to assess demographic influences. Marital status (unmarried/married), religion (Hindu/non-Hindu), family type (nuclear/joint), and living arrangements (single parent/both parents/joint family) were included to understand family dynamics. Parental marital status (separated/together), number of siblings (one/two/three or more), parental education (basic education: grade 1–8/secondary education: grade 9–12/above secondary), and parental employment status (unemployed/self-employed/foreign employment/government worker) were included to examine family structure and socioeconomic factors. Household income (less than NPR 50,000/more than NPR 50,000), family disability (yes/no), family death in the last 12 months (yes/no), and academic performance (A–A+/B–B+/C+/below C+) were also collected. School attendance (less than 20 days/more than 20 days), extracurricular activity participation (yes/no), and social support (family/friends) were included to assess engagement and support systems. Behavioral factors such as smoking (Never smoked/ Smoked in the past), alcohol use (Never/consumed in past), drug abuse history (Never/consumed in past), and family history of mental health issues (yes/no) were recorded to evaluate behavioral and familial health influences. Additionally, childhood trauma history (yes/no) and access to mental health services (yes/no) were included to assess social–environmental and healthcare-related factors influencing mental health. These variables were selected to comprehensively capture factors associated with adolescent mental health, ensuring a robust analysis of potential determinants.

## Statistical analysis

All statistical analyses were conducted using IBM SPSS Statistics, version 26. Descriptive statistics, including frequencies and percentages, were used to summarize the distribution of both independent and dependent variables. To explore associations between variables, both bivariate and multivariable analyses were performed.

At the bivariate level, the chi-square test was applied to assess associations between categorical independent and dependent variables. Variables with a p-value ≤ 0.25 at a 95% confidence interval were selected for multivariable logistic regression analysis. Further, variables were selected based on their theoretical and clinical significance. These variables were then entered into multivariable logistic regression models to identify predictors for anxiety, depression, and suicidal ideation among adolescents. Three separate models were run for anxiety, depression, and suicidal ideation, respectively. Effect sizes were reported in adjusted odds ratio (aOR) and 95% confidence interval (CI). The relationship was statistically significant at $p < 0.05$. To strengthen methodological transparency, multicollinearity among independent variables was assessed using the Variance Inflation Factor (VIF); all VIF values were below 5, indicating acceptable collinearity. Model goodness-of-fit was evaluated using the Hosmer–Lemeshow test, and all three models demonstrated adequate fit ($p > 0.05$).

## Results

### Socio-demographic characteristics of adolescents

The study assessed anxiety, depression, and suicidal ideation among 304 adolescents in Parbat District. More than two-thirds of the respondents were female, and the mean age was 16.7 years (Table 1). Over four-fifths were higher secondary students, and the majority belonged to the Brahmin/Chhetri (60.2%) ethnic group. Most participants were unmarried (96.7%), Hindu (90.8%), and living in nuclear families (60.5%). More than half of the adolescents' fathers had

**Table 1. Socio-demographic characteristics of adolescents.**

| Variable | Frequency (n = 304) | Percentage (%) |
|---|---|---|
| **Class** | | |
| Secondary (9–10) | 44 | 14.5 |
| Higher secondary (11–12) | 260 | 85.5 |
| **Age** | | 16.67 ± 1.28 |
| **Sex** | 360 | |
| Male | 95 | 31.3 |
| Female | 209 | 68.8 |
| **Ethnicity** | | |
| Dalit | 52 | 17.1 |
| Janajati | 69 | 22.7 |
| Brahmin/Chhetri | 183 | 60.2 |
| **Marital Status of Participants** | | |
| Unmarried | 294 | 96.7 |
| Married | 10 | 3.3 |
| **Religion** | | |
| Hindu | 276 | 90.8 |
| Non-Hindu | 28 | 9.2 |
| **Type of Family** | | |
| Nuclear | 184 | 60.5 |
| Joint | 120 | 39.5 |
| **Living Arrangements** | | |
| Single parent | 61 | 20.1 |
| Both parents | 119 | 39.1 |
| Joint family | 59 | 19.4 |
| Other arrangements | 65 | 21.4 |
| **Marital Status of Parents** | | |
| Separated | 20 | 6.6 |
| Together | 284 | 93.4 |
| **Number of Siblings** | | |
| One | 82 | 27.0 |
| Two | 125 | 41.1 |
| More than 3 | 97 | 31.9 |
| **Father's Education Level** | | |
| Basic Education (Class 1–8) and below | 135 | 44.4 |
| Secondary Education (Class 9–12) and above | 169 | 55.6 |
| **Father's Employment Status** | | |
| Unemployed | 25 | 8.2 |
| Self-employed | 153 | 50.3 |
| Foreign employment | 97 | 31.9 |
| Government worker | 29 | 9.5 |
| **Mother's Education Level** | | |
| Basic Education (Class 1–8) and below | 162 | 53.3 |
| Secondary Education (Class 9–12) and above | 142 | 46.7 |
| **Mother's Employment Status** | | |
| Unemployed | 123 | 40.5 |
| Self-employed | 136 | 44.7 |

*(Continued)*

**Table 1.** (Continued)

| Variable | Frequency (n = 304) | Percentage (%) |
|---|---|---|
| Foreign employment | 24 | 7.9 |
| Government worker | 21 | 6.9 |
| **Household Income** | | |
| Less than NPR 50,000 | 160 | 52.6 |
| More than NPR 50,000 | 144 | 47.4 |
| **Any Disability in Family** | | |
| No | 286 | 94.1 |
| Yes | 18 | 5.9 |
| **Family Death in Last 12 Months** | | |
| No | 290 | 95.4 |
| Yes | 14 | 4.6 |
| **Academic Performance** | | |
| A/A+ | 94 | 30.9 |
| B/B+ | 158 | 52.0 |
| C/C+ | 43 | 14.1 |
| Below C/C+ | 9 | 3.0 |
| **School Attendance** | | |
| Less than 20 days | 14 | 4.6 |
| More than 20 days | 290 | 95.4 |
| **Extracurricular Activities** | | |
| No | 69 | 22.7 |
| Yes | 235 | 77.3 |
| **Social Support and Network** | | |
| Family network | 189 | 62.2 |
| Friends network | 115 | 37.8 |
| **Smoking History** | | |
| Never | 288 | 94.7 |
| Past | 16 | 5.3 |
| **Alcohol History** | | |
| Never | 294 | 96.7 |
| Past | 10 | 3.3 |
| **Drug Abuse History** | | |
| Never | 298 | 98.0 |
| Past | 6 | 2.0 |
| **History of Mental Health Issues** | | |
| No | 284 | 93.4 |
| Yes | 20 | 6.6 |
| **Family History of Mental Health Issues** | | |
| No | 298 | 98.0 |
| Yes | 6 | 2.0 |
| **Childhood Trauma History** | | |
| No | 277 | 91.1 |
| Yes | 27 | 8.9 |
| **Access to Mental Health Services** | | |
| No | 287 | 94.4 |
| Yes | 17 | 5.6 |

had secondary education and 46.7% of the adolescents' mothers had secondary education. Fathers were predominantly self-employed (50.3%), whereas mothers were either unemployed (40.5%) or engaged in small-scale self-employment (44.7%). Nearly half of the households earned less than NPR 50,000 per month.

Regarding school-related characteristics, 30.9% of students had an A/A+ grade, 95.4% attended school regularly (>20 days a month), and the majority participated in extracurricular activities (77.3%). Family was the primary source of social support for 62.2% of adolescents. Risk behaviors such as smoking, alcohol consumption and drug use were rare. In terms of mental health, less than one in ten adolescents reported a history of mental health issues or childhood trauma, and only a small proportion (5.6%) had ever accessed mental health services.

## Prevalence of anxiety, depression, and suicidal ideation among adolescents

Table 2 shows that 36.5% of adolescents experienced anxiety, 17.4% had depression, and 7.6% reported suicidal ideation. The majority, 63.5%, did not have anxiety, 82.6% were not depressed, and 92.4% did not experience suicidal thoughts. Anxiety was the most common mental health issue among the adolescents.

## Factors associated with anxiety among adolescents

Table 3 shows the factors associated with anxiety. The analysis identified several significant factors associated with anxiety in adolescents. Female adolescents were significantly more likely to experience anxiety compared to males (aOR: 2.59, [1.30 – 5.19], $p < 0.01$). Non-Hindu adolescents had a higher odds of developing anxiety compared to their Hindu counterparts (aOR: 6.63, [1.96 – 22.42], $p < 0.01$). Adolescents with lower scholastic achievements (C/C+) were found to have a lower odds for anxiety (aOR: 0.27, [0.09 – 0.81], $p < 0.05$) compared to those scoring A/A+. Additionally, adolescents with a friends' network as a source of social support, exhibited a significantly higher likelihood of experiencing anxiety (aOR: 2.96, [1.58 – 5.52], $p < 0.01$) compared to a family network.

In contrast, several factors were not found to be statistically significant in relation to anxiety. These include class (higher secondary), ethnicity, marital status of participants, type of family, living arrangements, marital status of parents, number of siblings, and the education level and employment status of both parents. Household income, disability in the family, family death in the last 12 months, school attendance, extracurricular activities, smoking, alcohol, and drug abuse history, as well as family history of mental health issues, did not show significant associations with anxiety. Furthermore, access to mental health services did not significantly impact the likelihood of anxiety among the adolescents in this study. Notably, it is worth highlighting that the magnitude of the female gender effect differed substantially between anxiety (aOR: 2.59) and depression (aOR: 5.65), representing a nearly 2.5-fold difference in risk ratios across the two conditions. This differential suggests that, while biological and social stressors place female adolescents at elevated risk for both conditions, the factors driving depression among females may be particularly potent. The mechanistic basis for this discrepancy is discussed further in the Discussion section.

## Factors associated with depression among adolescents

Table 4 depicts the factors associated with depression. In the analysis of factors associated with depression, several variables were found to be significantly related. Female adolescents were significantly more likely to experience depression

**Table 2. Prevalence of anxiety, depression, and suicidal ideation among adolescents.**

| Particular | Yes, n (%) | No, n (%) |
|---|---|---|
| Anxiety | 111(36.5%) | 193(63.5%) |
| Depression | 53(17.4%) | 251(82.6%) |
| Suicidal Ideation | 23(7.6%) | 281(92.4%) |

**Table 3. Factors associated with anxiety among adolescents.**

| Variables | Adjusted OR | 95% CI |
|---|---|---|
| **Class** | | |
| Secondary (9–10) | Ref | |
| Higher secondary (11–12) | 1.34 | [0.38 – 4.68] |
| **Age** | 1.43 | [1.02 – 1.98] * |
| **Sex** | | |
| Male | Ref | |
| Female | 2.59 | [1.30 – 5.19] *** |
| **Religion** | | |
| Hindu | Ref | |
| Non-Hindu | 6.62 | [1.96 – 22.42] *** |
| **Type of Family** | | |
| Nuclear | Ref | |
| Joint | 0.81 | [0.39 – 1.67] |
| **Mother's Employment Status** | | |
| Unemployed | Ref | |
| Self-employed | 0.95 | [0.50 – 1.79] |
| Foreign employment | 0.31 | [0.08 – 1.19] |
| Government worker | 0.52 | [0.16 – 1.76] |
| **Household Income** | | |
| Less than 50,000 | Ref | |
| More than 50,000 | 0.7 | [0.36 – 1.26] |
| **Academic Performance** | | |
| A/A+ | Ref | |
| B/B+ | 0.60 | [0.31 – 1.16] |
| C/C+ | 0.27 | [0.09 – 0.80] ** |
| Below C/C+ | 0.54 | [0.08 – 3.66] |
| **Extracurricular Activities** | | |
| No | Ref | |
| Yes | 1.96 | [0.92 – 4.17] |
| **Social Support and Network** | | |
| Family network | Ref | |
| Friends network | 2.96 | [1.58 – 5.52] *** |
| **Alcohol History** | | |
| Never | Ref | |
| Past | 2.14 | [0.27 – 16.73] |
| **History of Mental Health Issues** | | |
| No | Ref | |
| Yes | 2.61 | [0.81– 8.45] |
| **Childhood Trauma History** | | |
| No | Ref | |
| Yes | 2.27 | [0.64– 8.09] |
| **Access to Mental Health Services** | | |
| No | Ref | |
| Yes | 0.77 | [0.16 – 3.76] |

* p-value<0.05; ** p-value<0.01; *** p-value<0.001.

CI = Confidence Interval; OR = Odds Ratio.

**Table 4. Factors associated with depression among adolescents.**

| Variables | Adjusted OR | 95% CI |
|---|---|---|
| **Sex** | | |
| Male | Ref | |
| Female | 5.65 | [1.82 – 17.53] *** |
| **Religion** | | |
| Hindu | Ref | |
| Non-Hindu | 7.84 | [1.94– 31.85] *** |
| **Household Income** | | |
| Less than 50,000 | Ref | |
| More than 50,000 | 0.38 | [0.16 – 0.89] * |
| **Any Disability in Family** | | |
| No | Ref | |
| Yes | 0.65 | [0.11 – 3.77] |
| **Academic Performance** | | |
| A/A+ | Ref | |
| B/B+ | 0.53 | [0.22 – 1.22] |
| C/C+ | 0.12 | [0.02 – 0.69] ** |
| Below C/C+ | 3.66 | [0.46 – 28.75] |
| **School Attendance** | | |
| Less than 20 days | Ref | |
| More than 20 days | 0.09 | [0.01 – 0.82] * |
| **Extracurricular Activities** | | |
| No | Ref | |
| Yes | 1.05 | [0.38 – 2.87] |
| **Social Support and Network** | | |
| Family network | Ref | |
| Friends network | 1.22 | [0.53 – 2.84] |
| **Smoking History** | | |
| Never | Ref | |
| Past | 1.53 | [0.13 – 17.50] |
| **Alcohol History** | | |
| Never | Ref | |
| Past | 0.37 | [0.03 – 4.88] |
| **Drug Abuse History** | | |
| Never | Ref | |
| Past | 4.29 | [0.18 – 99.20] |
| **History of Mental Health Issues** | | |
| No | Ref | |
| Yes | 2.75 | [0.59 – 12.84] |
| **Family History of Mental Health Issues** | | |
| No | Ref | |
| Yes | 3.02 | [0.25 – 36.33] |
| **Childhood Trauma History** | | |
| No | Ref | |
| Yes | 15.61 | [3.18 – 76.67] *** |

*(Continued)*

**Table 4.** (Continued)

| Variables | Adjusted OR | 95% CI |
|---|---|---|
| **Access to Mental Health Services** | | |
| No | Ref | |
| Yes | 0.47 | [0.06 – 3.73] |

\* p-value < 0.05; \*\* p-value < 0.01; \*\*\* p-value < 0.001.

CI = Confidence Interval; OR = Odds Ratio.

compared to males (aOR: 5.65, [1.82 – 17.53], p < 0.001). Non-Hindu adolescents also had had a higher odds for depression (aOR: 7.86, [1.94 – 31.85], p < 0.01). Adolescents with lower academic performance (C/C+) were found to have a significantly lower likelihood of depression (aOR: 0.13, [0.02 – 0.70], p < 0.01). Furthermore, those who had attended school for more than 20 days were significantly less likely to experience depression (aOR: 0.09, [0.01 – 0.82], p < 0.05). A history of childhood trauma was another significant factor associated with depression (aOR: 15.61, [3.18 – 76.67], p < 0.001).

Several other variables were also found to be non-significant in relation to depression. These included class (higher secondary), age, ethnicity, marital status of participants, family type, marital status of parents, number of siblings, and the education and employment status of both parents. Household income (more than 50,000), disability in the family, family death in the last 12 months, academic performance (B/B+), extracurricular activities, social support networks, and smoking, alcohol, and drug abuse history did not show significant associations with depression. Additionally, having a family history of mental health issues and access to mental health services were not significantly related to depression in this study.

### Factors associated with suicidal ideation among adolescents

Table 5 revealed several significant factors associated with suicidal ideation among adolescents. Higher secondary education (11–12) was significantly linked to a lower likelihood of suicidal ideation (aOR: 0.05, [0.00 – 0.79], p < 0.05). Age was significant factor as one year increase in age was associated with 2.23 times higher odds of suicidal ideation (aOR: 2.23, [1.05–4.76], p < 0.05). A family member's death in the past 12 months was strongly associated with increased odds of suicidal ideation (aOR: 10.67, [1.05–108.64], p < 0.05). Additionally, having a supportive friends network significantly decreased the likelihood of suicidal ideation (aOR: 0.09, [0.01–0.52], p < 0.05). A history of mental health issues was found to substantially increase the risk of suicidal ideation (aOR: 38.83, [4.71–320.26], p < 0.001), and a history of childhood trauma also emerged as a significant factor (aOR: 16.14, [1.89–138.15], p < 0.01).

In contrast, several factors were not significantly associated with suicidal ideation. These include sex, ethnicity, marital status, religion, type of family, living arrangements, marital status of parents, number of siblings, and the education and employment status of parents. Additionally, household income, the presence of disability in the family, school attendance, participation in extracurricular activities, and smoking, alcohol, or drug use history were not significant predictors of suicidal ideation. Furthermore, family history of mental health issues and access to mental health services did not show significant associations with suicidal ideation in the current study.

### Discussion

Anxiety, depression, and suicidal ideation are deeply interconnected mental health challenges that affect adolescents worldwide. This study aimed to investigate the prevalence of anxiety, depression, and suicidal ideation among adolescents, alongside their associated risk and protective factors. Our findings reveal the pervasive influence of gender, academic performance, and childhood adversity or prior mental health issues across all three conditions. These results are

**Table 5. Factors associated with suicidal ideation among adolescents.**

| Variables | Adjusted OR | 95% C. I. |
|---|---|---|
| **Class** | | |
| Secondary (9–10) | Ref | |
| Higher secondary (11–12) | 0.04 | 0.00 – 0.79 * |
| **Age** | 2.23 | 1.04– 4.76 * |
| **Sex** | | |
| Male | Ref | |
| Female | 0.51 | 0.12 – 2.06 |
| **Religion** | | |
| Hindu | Ref | |
| Non-Hindu | 2.78 | 0.25– 30.91 |
| **Any Disability in Family** | | |
| No | Ref | |
| Yes | 0.96 | 0.12 – 7.80 |
| **Family Death in Last 12 Months** | | |
| No | Ref | |
| Yes | 10.67 | 1.04 – 108.63 * |
| **Extracurricular Activities** | | |
| No | Ref | |
| Yes | 0.46 | 0.11 – 1.90 |
| **Social Support and Network** | | |
| Family network | Ref | |
| Friends network | 0.08 | 0.01 – 0.52 *** |
| **Smoking History** | | |
| Never | Ref | |
| Past | 5.37 | 0.45 – 63.09 |
| **Alcohol History** | | |
| Never | Ref | |
| Past | 0.92 | 0.05 – 16.77 |
| **Drug Abuse History** | | |
| Never | Ref | |
| Past | 4.18 | 0.12– 140.41 |
| **History of Mental Health Issues** | | |
| No | Ref | |
| Yes | 38.82 | 4.70 – 320.25*** |
| **Childhood Trauma History** | | |
| No | Ref | |
| Yes | 16.13 | 1.88 – 138.14** |
| **Access to Mental Health Services** | | |
| No | Ref | |
| Yes | 1.41 | 0.17 – 11.48 |

* p-value < 0.05; ** p-value < 0.01; *** p-value < 0.001.

CI = Confidence Interval; OR = Odds Ratio.

largely consistent with broader national and global trends, underscoring the urgent need for targeted and comprehensive mental health interventions for young people [29].

The study revealed the prevalence of anxiety to be 36.5% in the Parbat district, in line with the survey conducted by Pokharel et al. [30] Similar studies showed a higher prevalence of anxiety among adolescents ranging from 45% to 55% approximately [11,22]. This figure is notably higher than the overall pooled global prevalence of anxiety reported among adolescents aged 12–17 years (9.0%) from a population-based study across 82 countries between 2003 and 2015 [31]. Likewise, the prevalence of anxiety symptoms reported by the current study was higher than that of the study conducted in China, Sri Lanka, and Vietnam, but lower than in India [32–35]. These differences in the prevalence of anxiety symptoms observed in these studies can be partially attributed to variations in sample sizes, study populations, culture, and assessment methods.

This study revealed several significant factors associated with anxiety. Female adolescents are more likely to experience anxiety symptoms than male adolescents, a finding consistently supported by existing literature from Nepal and globally [22,30,33,36]. The possible explanation for this is that females are subjected to more stressful conditions like sexual and domestic violence, which makes them prone to anxiety symptoms. Likewise, the female adolescent stage is marked by excessive hormonal changes [37]. This highlights the need for more tailored mental health interventions for female adolescents, focusing on stressors such as societal roles and violence. Furthermore, lower academic performance was significantly linked to a higher odds of anxiety. This aligns with previous research indicating that academic struggles can exacerbate anxiety levels in adolescents [38,39]. [38] Adolescents struggling academically often face increased parental expectations, social stigma, and reduced self-esteem, exacerbating their anxiety levels [40–42]. These findings put forward the need for mentorship, tutoring, and counseling services for students with low performance, aiming to reduce academic stress and foster a supportive environment for adolescents. Similarly, a significant association was seen between a history of childhood trauma and anxiety, which is corroborated by previous studies [43,44]. This showed the need for trauma-informed care, academic support, and gender-sensitive mental health interventions for adolescents. Moreover, addressing the long-term impacts of early trauma through treatment and support systems is crucial for reducing anxiety and enhancing general well-being. The elevated likelihood of anxiety among non-Hindu adolescents underscores the possible impact of cultural and religious identity on mental health outcomes. This aligns with the previous study, which revealed cultural and religious connections can influence coping methods, community support, and adaptability to stressors [45–47].

Interestingly, while friends' networks were linked to anxiety symptoms in this study, however, previous research suggests that family networks often provide more consistent emotional support, which can be a protective factor against anxiety [48–50]. [50,51] A history of mental health issues also emerged as a significant predictor of anxiety. This agrees with longitudinal research showing that previous mental health issues predispose people to future episodes of anxiety and other psychiatric diseases [52]. The combination of unresolved psychological distress, higher sensitivity to stimuli, and potential biological vulnerabilities all contribute to this elevated risk. Early life trauma, neglect, and ongoing stress are significant environmental contributors to the development of anxiety disorders during adolescence [53]. This research emphasizes the vital importance of early detection and long-term mental health monitoring, especially among adolescents with prior mental health issues [53].

Depression in adolescents shares multiple risk factors with anxiety but presents unique associations that require further examination. The observed prevalence of depression in this study (17.4%) aligns closely with findings from adolescent populations worldwide, affirming that adolescent depression is a critical public health concern [54,55]. This rate is consistent with previous studies in diverse settings, where adolescent depression prevalence ranges from 6.4% to 52.9%, highlighting a significant variability in the existing literature across different regions and contexts [9,10,56,57]. The results identified several factors associated with adolescent depression, including sex, religion, academic performance, school attendance, and a history of childhood trauma.

Similar to anxiety, female adolescents were found to have a significantly higher risk of depression, a finding consistent with global studies [58,59]. This susceptibility of female adolescents with depression might be due to a combination of biological and psychological factors, with hormonal changes during puberty, alongside stressors such as body image concerns and social pressures, contributing to this disparity [60]. Likewise, Non-Hindu adolescents faced higher risk of depression, potentially due to the unique socio-cultural challenges like social exclusion or discrimination that minority groups face [61]. Additionally, religious affiliation impacts social support networks, which are essential for the mental well-being of adolescents [62,63]. Academic performance emerged as another important factor, with lower-performing students (grades C/C+) displaying a higher likelihood of depression. These findings are corroborated by research among adolescent students in public schools in Kathmandu, which identified academic pressure as a predictor of depression [64]. Academic struggles coupled with lower self-esteem, increased parental expectations, and social stigma, exacerbate depressive symptoms [64,65].

Interestingly, students attending school for fewer than 20 days showed a reduced likelihood of depression. Similar studies showed school absenteeism as a significant contributor to depressive symptoms among adolescents [65,66]. While this result may seem counterintuitive, it could reflect cultural or contextual nuances, where absenteeism is either a coping mechanism to avoid stress or a sign of disengagement from school-related pressures that may lead to depression. Alternatively, adolescents with depression might find it challenging to attend school regularly, which further contributes to isolation and social withdrawal. However, this finding warrants cautious interpretation. It may reflect survival bias, whereby only the most resilient or less severely affected students with low attendance remained enrolled and were captured in the sample, while more severely depressed adolescents had already dropped out of school entirely and were therefore excluded from the study. Population heterogeneity represents another plausible explanation: students with very low attendance may include those engaged in alternative educational settings, vocational training, or part-time work, who may experience fundamentally different stressors than classroom-based peers. Furthermore, the cross-sectional design precludes any conclusions about the direction of effect between attendance and depression. The small number of participants in the low-attendance category (n = 14) also limits statistical precision and contributes to the wide confidence interval observed (aOR: 0.09; 95% CI: 0.01–0.82). Sensitivity analyses using an alternative attendance cutoff (e.g., fewer than 30 days) were not feasible given the limited cell sizes in this subgroup; future studies with larger samples should explicitly test the stability of this association across different attendance thresholds to determine whether it represents a genuine protective mechanism, survival bias, or unmeasured confounding [67]. A history of childhood trauma was one of the strongest predictors of depression in this study. Trauma in early life is a well-documented risk factor for mental health issues, including depression, due to its lasting impact on emotional regulation and stress response systems [68]. Previous study indicated that traumatic experiences can alter brain development and heighten stress responses, increasing vulnerability to depression in adolescence [49]. These results emphasize the need for comprehensive approaches that address academic support, cultural sensitivity, and the long-term effects of trauma.

Suicidal ideation in adolescents is presented as a pressing mental health concern, often rooted in untreated anxiety and depression. The study found 7.6% of adolescents reported suicidal ideation, which is lower than global prevalence, i.e., 14% as reported in Global School-based Student Health Survey (GSHS) data [31] and lower than findings of a study conducted by Pandey et al. [6] However, higher than studies conducted in different countries [69,70]. The study also found that older adolescents exhibited a higher likelihood of experiencing suicidal thoughts. This is in line with the existing studies that revealed increasing prevalence of suicidal ideation among later adolescents [71–73]. Similarly, poor academic performance was another significant predictor of suicidal ideation, aligning with previous research that has linked educational struggles to increased mental distress and hopelessness [74,75].

Unlike anxiety, where reliance on peer networks was a risk factor, adolescents with strong friendships showed a lower likelihood of suicidal ideation. Similar findings are suggested by the findings from the Global School-based Health Survey in Nepal [6]. Additionally, prior mental health issues emerged as strong predictors of suicidal ideation, reinforcing the

long-term consequences of early-life adversity. These findings are in line with a study using data from the Global School-based Student Health Survey in Nepal, which identified factors such as anxiety, and loneliness as risk factors for suicidal ideation among adolescents [6]. These findings were also supported by the study conducted in Canada, which linked the occurrence of anxiety and depressive symptoms to suicidal ideation [76]. Similarly, the history of childhood trauma showed a significant association with suicidal ideation. These findings were supported by existing literature which highlighted the relationship between experiences of childhood trauma and suicidal ideation [77–79]. The emotional turmoil and traumatic experiences among adolescents might lead to negative self-concepts, feelings of worthlessness, lack of healthy coping mechanisms, and social isolation, which ultimately contribute to suicidal thoughts. These findings underscore the importance of school-based mental health programs, grief counseling, and peer-support initiatives in suicide prevention efforts.

Addressing the causes of anxiety, depression, and suicidal ideation in adolescents requires comprehensive interventions. Early identification and treatment are crucial, involving strategies such as cognitive-behavioral therapy (CBT), which has been effective in managing anxiety and depression by helping adolescents develop coping skills and challenge negative thought patterns [80]. The Substance Abuse and Mental Health Services Administration (SAMHSA) provides guides on effective treatments for suicidal thoughts and behaviors among youth, emphasizing the importance of accessible mental health services [81]. Additionally, fostering supportive environments at home and school, promoting healthy peer relationships, and ensuring access to mental health resources can significantly reduce the prevalence of these mental health issues among adolescents [82,83]. Holistic, culturally sensitive approaches in school and community mental health programs can be instrumental in reducing mental health risks and supporting the mental well-being of adolescents. Most importantly, understanding the contributing factors and implementing targeted interventions are essential steps toward mitigating these issues and promoting the well-being of young individuals [84–87].

### Strengths

This study has several notable strengths. It is the first to assess anxiety, depression, and suicidal ideation among adolescents in the under-researched Parbat District of Nepal, addressing a critical gap in rural mental health data. Methodologically, it employed the internationally validated tools (GAD-7, PHQ-9, SBQ-R) translated into Nepali, demonstrating high internal consistency (Cronbach's $\alpha > 0.92$). Representative a proportionate and systematic random sampling across 10 schools ensured socioeconomic diversity. Comprehensive multivariable analyses identified key determinants- gender, academic performance, trauma history, and social networks- providing nuanced insights into shared and unique risk factors. Ethical standards were rigorously maintained with the NHRC approval, strict consent/assent procedures, and confidentiality safeguards, and a high response rate (90%) minimized non-response bias.

### Limitations

Despite these strengths, the study has several limitations. Its cross-sectional design prevents causal inference, and the sample was limited to grades 11–12 (ages 16–19) within Kushma Municipality, restricting generalizability and excluding early adolescents. Therefore, even if causal language is used inadvertently or causal linking terms are applied, the findings should be interpreted as correlational rather than indicative of a cause-and-effect relationship. While the results provide valuable insights for policymakers, schools, and communities, the findings are generalizable to adolescents aged 16–19 years enrolled in grades 11–12 within Kushma Municipality. They should not be extrapolated to younger adolescents (ages 10–15), adolescents outside the district, or youth who are out of formal schooling. Self-reporting may have led to underreporting of sensitive issues such as trauma or substance use, and key confounders like bullying, social media exposure, and detailed family dynamics were not assessed. Additionally, some odds ratios had wide confidence intervals due to small subgroup sizes, reducing precision.

## Study implications

Findings highlight an urgent need for school-based mental health interventions, gender-sensitive support programs, trauma-informed care, and culturally inclusive initiatives. Policymakers should prioritize adolescent mental health through targeted interventions, parental engagement, and investment in rural mental health infrastructure. As adverse childhood experiences and earlier mental health issues are strongly predicting mental health in adolescents, it is very important to act on a household level during the childhood period rather than in the adolescent period. A parental education program that teaches non-violent discipline, emotional regulation, and a positive parent-child relationship could be promoted. Likewise, screening for adversity in school or healthcare settings could help children referred to primary care and get services early. Likewise, the school-based mental health programs, such as psychoeducation, counseling, and coping skill training, could be implemented, and female students should be prioritized as well. Future research should include longitudinal and mixed methods design and include early adolescents and cover larger areas of Nepal to improve generalizability.

## Conclusion

This study reveals high rates of anxiety (36.5%), depression (17.4%), and suicidal ideation (7.6%) among adolescents in Parbat, Nepal. Key predictors include female gender, non-Hindu religious status, academic performance, history of childhood trauma, and prior mental health treatment. These findings highlight the urgent need for targeted mental health interventions prioritizing these high-risk groups.

## Supporting information

**S1 Data. Supporting dataset used for the analysis presented in this study.**
(PDF)

## Acknowledgments

We would like to express our sincere gratitude to all individuals who were directly and indirectly involved in this research. We thank the Nepal Health Research Council (NHRC) for providing ethical clearance, and Kushma Municipality, Education Section, for providing the list of schools within the municipality. We are also grateful to the principals and teachers at the participating schools for their support in facilitating the study. Finally, we deeply appreciate the valuable participation of the study participants.

## Author contributions

**Conceptualization:** Anjali P.C, Prabin Sharma, Suman Sharma.

**Data curation:** Anjali P.C, Rajan Bhusal.

**Formal analysis:** Rajan Bhusal, Shreesti Sharma.

**Methodology:** Anjali P.C.

**Project administration:** Anjali P.C.

**Resources:** Anjali P.C.

**Supervision:** Anjali P.C.

**Validation:** Anjali P.C, Rajan Bhusal, Krishna Prasad Sapkota.

**Visualization:** Anjali P.C, Krishna Prasad Sapkota.

**Writing – original draft:** Anjali P.C, Rajan Bhusal, Anjali Bhatt, Shreesti Sharma.

**Writing – review & editing:** Anjali P.C, Rajan Bhusal, Krishna Prasad Sapkota, Prabin Sharma.

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
