## [Decision Letter · Decision Letter 0]

17 Mar 2026

PMEN-D-26-00096

PREVALENCE AND DETERMINANTS OF ANXIETY, DEPRESSION, AND SUICIDAL IDEATION AMONG ADOLESCENTS OF PARBAT DISTRICT: A CROSS-SECTIONAL STUDY

PLOS Mental Health

Dear Dr. Anjali P.C,

Thank you for submitting your manuscript to PLOS Mental Health. After careful consideration, we feel that it has merit but does not fully meet PLOS Mental Health’s publication criteria as it currently stands. Therefore, we invite you to submit a revised version of the manuscript that addresses the points raised during the review process.

Please revise your manuscript in accordance with the reviewers' suggestions.

We look forward to receiving your revised manuscript.

Kind regards,

Wenxi Sun, M.D.

Academic Editor

PLOS Mental Health

**Journal Requirements:**

1. Please include a complete copy of PLOS’ questionnaire on inclusivity in global research in your revised manuscript. Our policy for research in this area aims to improve transparency in the reporting of research performed outside of researchers’ own country or community. The policy applies to researchers who have travelled to a different country to conduct research, research with Indigenous populations or their lands, and research on cultural artefacts. The questionnaire can also be requested at the journal’s discretion for any other submissions, even if these conditions are not met.  Please find more information on the policy and a link to download a blank copy of the questionnaire here: https://journals.plos.org/mentalhealth/s/best-practices-in-research-reporting. Please upload a completed version of your questionnaire as Supporting Information when you resubmit your manuscript.

2. Please ensure that your Ethics Statement is available in its entirety at the beginning of your Methods section, under a subheading 'Ethics Statement'.

3. Please upload separate figure files in .tif or .eps format. Also, remove the figures from your manuscript file but keep the legends.

https://journals.plos.org/mentalhealth/s/figures

https://journals.plos.org/mentalhealth/s/figures#loc-file-requirements

4. In the online submission form, you indicated that “The dataset used and analyzed during the study are available from the corresponding author on reasonable request.”.

3. Uploaded as supplementary information.

5. Some material included in your submission may be copyrighted. According to PLOS’s copyright policy, authors who use figures or other material (e.g., graphics, clipart, maps) from another author or copyright holder must demonstrate or obtain permission to publish this material under the Creative Commons Attribution 4.0 International (CC BY 4.0) License used by PLOS journals. Please closely review the details of PLOS’s copyright requirements here: PLOS Licenses and Copyright. If you need to request permissions from a copyright holder, you may use PLOS's Copyright Content Permission form.

Potential Copyright Issues:

a. Figure 1: please (a) provide a direct link to the base layer of the map (i.e., the country or region border shape) and ensure this is also included in the figure legend; and (b) provide a link to the terms of use / license information for the base layer image or shapefile. We cannot publish proprietary or copyrighted maps (e.g. Google Maps, Mapquest) and the terms of use for your map base layer must be compatible with our CC-BY 4.0 license.

**Additional Editor Comments (if provided):**

Reviewers' comments:

Reviewer's Responses to Questions

**Comments to the Author**

1. Does this manuscript meet PLOS Mental Health’s publication criteria? Is the manuscript technically sound, and do the data support the conclusions? The manuscript must describe methodologically and ethically rigorous research with conclusions that are appropriately drawn based on the data presented.

Reviewer #1: Yes

Reviewer #2: Partly

2. Has the statistical analysis been performed appropriately and rigorously?

Reviewer #1: Yes

Reviewer #2: N/A

3. Have the authors made all data underlying the findings in their manuscript fully available (please refer to the Data Availability Statement at the start of the manuscript PDF file)?

Reviewer #1: Yes

Reviewer #2: Yes

4. Is the manuscript presented in an intelligible fashion and written in standard English?

Reviewer #1: Yes

Reviewer #2: No

5. Review Comments to the Author

**Reviewer #1:** This is a well-conducted cross-sectional study investigating the prevalence and associated factors of anxiety, depression, and suicidal ideation among high school adolescents in Parbat District, Nepal. The study addresses a critical public health gap in a rural, under-researched population. Methodologically sound with rigorous ethical standards, internationally validated instruments with excellent internal consistency (Cronbach's α > 0.92), and representative sampling across multiple schools, the paper provides valuable epidemiological data and identifies key risk factors. The findings are contextualized appropriately against national and global evidence. Overall, I recommend acceptance for publication with minor revisions to strengthen specific sections.

Abstract

The abstract is well-structured and comprehensively summarizes the study's background, methods, results, and conclusions. To enhance clarity and impact, I recommend that you emphasize the most striking finding in the results section: the extremely strong association between prior mental health history and suicidal ideation (aOR: 38.83; 95% CI: 4.71–320.26), which substantially exceeds the magnitude of other risk factors. Highlighting this would immediately draw readers' attention to the study's most clinically significant discovery.

Introduction

The introduction effectively contextualizes the problem using both global and Nepal-specific epidemiological data. However, to strengthen the theoretical foundation, I recommend you expand the discussion of why women experience higher rates of anxiety versus depression. Beyond mentioning hormonal changes and exposure to sexual violence, consider briefly incorporating social-psychological mechanisms such as gender role socialization and differential stress appraisal patterns, as these may explain why the female odds ratio for depression (aOR: 5.65) is substantially higher than for anxiety (aOR: 2.59). This deeper mechanistic insight would provide better scaffolding for interpreting your results.

Methods

The methodological approach is sound and well-documented. One clarification would strengthen the sample size calculation: explicitly state the source literature or prior study (reference 23 appears relevant) from which the baseline prevalence of 40.6% was derived, and briefly justify why this figure is appropriate for the Parbat population. Additionally, although the pretest was conducted in Phalewas Municipality while the main study occurred in Kushma Municipality (both within Parbat District), a brief note on whether the two municipalities differ substantially in socioeconomic or demographic characteristics would address potential concerns about whether the pretest findings generalize to the main study population.

Results

The results are presented clearly with well-organized tables and logically structured findings. However, one inconsistency deserves mention: the magnitude of the female gender effect differs substantially between anxiety (aOR: 2.59) and depression (aOR: 5.65). While both are significant, the paper would benefit from a brief explanatory statement in the results section noting this nearly 2.5-fold difference in risk ratios across the two conditions, preparing readers for the more detailed mechanistic discussion in the Discussion section.

Figures and Tables

The tables are formatted appropriately and include necessary confidence intervals and p-values. I recommend adding a participant flow diagram (formatted according to STROBE guidelines) showing the number of students enrolled, those completing questionnaires, and any exclusions or missing data. Additionally, you might consider generating a forest plot displaying the adjusted odds ratios and their 95% confidence intervals for the three conditions side-by-side, which would allow readers to visually compare risk magnitudes across mental health outcomes and independent variables.

Discussion

The discussion is comprehensive and appropriately contextualizes findings against existing literature. However, the interpretation of one finding—the protective effect of low school attendance (<20 days) against depression (aOR: 0.09; 95% CI: 0.01–0.82)—remains somewhat speculative. While you acknowledge the possibility of reverse causality (depressed adolescents missing more school), you do not sufficiently explore whether unmeasured confounders (e.g., students in alternative education settings, those working part-time) might explain this counterintuitive association. I recommend strengthening this section by explicitly acknowledging that this finding may reflect survival bias or population heterogeneity rather than a genuine protective mechanism of absenteeism. Furthermore, address whether sensitivity analyses using different school attendance cutoffs (e.g., <30 days) were performed to test the stability of this association.

Conclusions

The conclusion accurately reflects the study's findings and appropriately identifies the urgent need for targeted mental health interventions. To enhance practical utility, I recommend that you be more specific about which groups require prioritized intervention: explicitly name female adolescents, non-Hindu religious minorities, students with documented childhood trauma, and those with prior mental health treatment history as distinct high-risk populations. Additionally, clarify the generalizability constraints: the study sample comprises only grades 11–12 (ages 16–19) within Kushma Municipality, and findings should not be extrapolated to early adolescents (ages 10–15), rural areas outside the district, or youth who have left formal schooling.

Minor Suggestions

1. Line 385–390: The discussion of why friends' networks predict higher anxiety (contrary to a protective effect on suicidal ideation) is interesting but underdeveloped. Consider whether peer networks in this context may involve anxiety-provoking social comparison or exposure to secondary trauma rather than protective support.

2. Verify consistency of in-text citations and reference formatting throughout the manuscript.

3. Consider explicitly stating that the study meets STROBE guidelines for cross-sectional studies to enhance transparency and quality perception.

**Reviewer #2:** This manuscript examines the prevalence and associated factors of anxiety, depression, and suicidal ideation among high school adolescents in Nepal, using a cross-sectional study design. The topic is important, as adolescent mental health is a growing global public health concern. The use of standardized screening instruments and multivariable logistic regression analysis is appropriate for the research objectives.

Overall, the study contributes valuable information regarding mental health challenges among adolescents in rural Nepal. However, several methodological clarifications, improvements in reporting, and stronger interpretation of findings are required before the manuscript can be considered for publication.

Major Comments:

1. Sampling Procedure and Representativeness

Although the study mentions systematic and proportionate sampling across schools, more detail is needed regarding how schools and participants were selected. The authors should clarify whether the sampling approach ensures representativeness of adolescents in the district. Additionally, the inclusion of only school-going adolescents may limit the generalizability of the findings to adolescents who are not enrolled in school.

2. Sample Size Justification

The manuscript would benefit from a clearer explanation of the sample size calculation, including the assumptions used (expected prevalence, confidence level, margin of error). This information is important for evaluating whether the sample size is adequate to detect meaningful associations.

3. Logistic Regression Modeling

The authors report using multivariable logistic regression models to identify factors associated with anxiety, depression, and suicidal ideation. However, additional details regarding the model-building process would strengthen the methodological transparency. Specifically, the authors should clarify:

Whether multicollinearity among independent variables was assessed

Whether model goodness-of-fit statistics were evaluated

The criteria used to select variables for the final models.

4. Counterintuitive Association with School Attendance

The analysis suggests that poor school attendance is associated with lower odds of depression. This finding appears counterintuitive and may reflect issues related to variable coding, confounding, or measurement. The authors should clarify how school attendance was categorized and discuss potential explanations for this result.

5. Figures

The results section would benefit from the inclusion of graphical visualizations (e.g., bar charts for prevalence estimates or forest plots for adjusted odds ratios). These figures would improve the readability and interpretation of the findings.

Minor Comments:

1. Language and Grammar

Several sentences require minor grammatical revision and language polishing. Professional language editing is recommended to improve readability and clarity.

2. Redundancy in the Background Section

Some parts of the background section repeat similar information regarding the global burden of adolescent mental health problems. The section could be slightly condensed to improve focus and flow.

3. Consistency of Terminology

The manuscript alternates between terms such as “mental health issues,” “mental health problems,” and “mental disorders.” Using consistent terminology throughout the manuscript would improve clarity.

4. Tables

Tables should include clearer labeling of variables and categories. Providing sample counts (n) alongside percentages would help readers better interpret the data.

Despite the issues noted above, the study has several important strengths:

Use of validated screening tools for assessing anxiety, depression, and suicidal ideation.

Inclusion of multiple potential risk factors such as academic performance, trauma history, and social support.

Focus on a rural population in a low-resource setting, which is often underrepresented in mental health research.

Application of multivariable statistical analysis to explore associated factors.

6. PLOS authors have the option to publish the peer review history of their article (what does this mean?). If published, this will include your full peer review and any attached files.

**Do you want your identity to be public for this peer review?** For information about this choice, including consent withdrawal, please see our Privacy Policy.

Reviewer #1: No

Reviewer #2: No

Figure Resubmissions:

---

## [Decision Letter · Decision Letter 1]

22 Apr 2026

PREVALENCE AND DETERMINANTS OF ANXIETY, DEPRESSION, AND SUICIDAL IDEATION AMONG ADOLESCENTS OF PARBAT DISTRICT: A CROSS-SECTIONAL STUDY

PMEN-D-26-00096R1

Dear Anjali P.C,

We are pleased to inform you that your manuscript 'PREVALENCE AND DETERMINANTS OF ANXIETY, DEPRESSION, AND SUICIDAL IDEATION AMONG ADOLESCENTS OF PARBAT DISTRICT: A CROSS-SECTIONAL STUDY' has been provisionally accepted for publication in PLOS Mental Health.

Best regards,

Wenxi Sun, M.D.

Academic Editor

PLOS Mental Health

Reviewer Comments (if any, and for reference):

Reviewer's Responses to Questions

**Comments to the Author**

1. If the authors have adequately addressed your comments raised in a previous round of review and you feel that this manuscript is now acceptable for publication, you may indicate that here to bypass the “Comments to the Author” section, enter your conflict of interest statement in the “Confidential to Editor” section, and submit your "Accept" recommendation.

Reviewer #1: All comments have been addressed

2. Does this manuscript meet PLOS Mental Health’s publication criteria? Is the manuscript technically sound, and do the data support the conclusions? The manuscript must describe methodologically and ethically rigorous research with conclusions that are appropriately drawn based on the data presented.

Reviewer #1: Yes

3. Has the statistical analysis been performed appropriately and rigorously?

Reviewer #1: Yes

4. Have the authors made all data underlying the findings in their manuscript fully available (please refer to the Data Availability Statement at the start of the manuscript PDF file)?

Reviewer #1: Yes

5. Is the manuscript presented in an intelligible fashion and written in standard English?

Reviewer #1: Yes

6. Review Comments to the Author

Reviewer #1: References 16 and 17 are displaying as garbled text. After this modification, it can be accepted.

7. PLOS authors have the option to publish the peer review history of their article (what does this mean?). If published, this will include your full peer review and any attached files.

**Do you want your identity to be public for this peer review?** For information about this choice, including consent withdrawal, please see our Privacy Policy.

Reviewer #1: No
